# Machine Learning Models for the Automatic Detection of Exercise Thresholds in Cardiopulmonary Exercising Tests: From Regression to Generation to Explanation

**DOI:** 10.3390/s23020826

**Published:** 2023-01-11

**Authors:** Andrea Zignoli

**Affiliations:** Department of Industrial Engineering, University of Trento, 38123 Trento, Italy; andrea.zignoli@unitn.it

**Keywords:** artificial intelligence, machine learning, deep learning, cardiopulmonary stress test, explainable AI, machine learning interpretability

## Abstract

The cardiopulmonary exercise test (CPET) constitutes a gold standard for the assessment of an individual’s cardiovascular fitness. A trend is emerging for the development of new machine-learning techniques applied to the automatic process of CPET data. Some of these focus on the precise task of detecting the exercise thresholds, which represent important physiological parameters. Three are the major challenges tackled by this contribution: (A) regression (i.e., the process of correctly identifying the exercise intensity domains and their crossing points); (B) generation (i.e., the process of artificially creating a CPET data file ex-novo); and (C) explanation (i.e., proving an interpretable explanation about the output of the machine learning model). The following methods were used for each challenge: (A) a convolutional neural network adapted for multi-variable time series; (B) a conditional generative adversarial neural network; and (C) visual explanations and calculations of model decisions have been conducted using cooperative game theory (Shapley’s values). The results for the regression, generation, and explanatory techniques for AI-assisted CPET interpretation are presented here in a unique framework for the first time: (A) machine learning techniques reported an expert-level accuracy in the classification of exercise intensity domains; (B) experts are not able to substantially differentiate between a real vs an artificially generated CPET; and (C) Shapley’s values can provide an explanation about the choices of the algorithms in terms of ventilatory variables. With the aim to increase their technology-readiness level, all the models discussed in this contribution have been incorporated into a free-to-use Python package called pyoxynet (ver. 12.1). This contribution should therefore be of interest to major players operating in the CPET device market and engineering.

## 1. Introduction

### 1.1. Background

Cardiopulmonary exercise testing (CPET) is extensively adopted in the assessment of an individual’s cardiovascular and ventilatory systems [1]. During a CPET, the individual exercises on a treadmill or on a cycle ergometer according to a standardized protocol, with progressive increases in the exercise workload (the speed and elevation of the treadmill or the power output of the ergometer) until volitional exhaustion [2]. A metabolic cart is used together with a face mask to measure the flow and the O_2_ and CO_2_ concentration of the gasses exchanged at the airways, which are tightly coupled with cardiac output, pulmonary blood circulation and peripheral O_2_ extraction [3].

The following variables are available with a CPET: O_2_ uptake (VO_2_, mlO_2_/min), exhaled CO_2_ (VCO_2_, mlCO_2_/min), ventilation (VE, l/min), respiratory frequency (Rf, bpm), end-tidal O_2_ (PetO_2_, mmHg) and CO_2_ (PetCO_2_, mmHg), heart rate (HR, bpm) and working load (measured in W in cycling or speed and inclination in treadmill running). In Figure 1, a CPET of a representative individual is provided.

In normal conditions, three different patterns of CPET variables should be discernible after a well-conducted CPET. These patterns identify exercise intensities (or domains), and their corresponding metabolic conditions. Here, their description was inspired by the work of Keir et al. [4]:Pattern 1 (moderate intensity domain) is characterized by: (a) increasing VO_2_ and VCO_2_; (b) increasing VE; (c) decreasing ventilatory equivalents (i.e., VEVO_2_ and VEVCO_2_, computed as VE/VO_2_ and VE/VCO_2_); (d) decreasing PetO_2_ and increasing PetCO_2._Pattern 2 (heavy intensity domain) is characterized by: (a) increasing VO_2_ and VCO_2_; (b) increasing PetO_2_ and steady PetCO_2_; (c) increasing VEVO_2_ and steady VEVCO_2_; (d) increasing VE (with a slope greater than in Pattern 1).Pattern 3 (severe intensity domain) is characterized by: (a) increasing VO_2_ and VCO_2_; (b) increasing PetO_2_ (with a slope greater than in Pattern 2) and decreasing PetCO_2_; (c) increasing VEVO_2_ (with a slope greater than in Pattern 2) and VEVCO_2_; (d) increasing VE (with a slope greater than in Pattern 2).

Given that the workload during a CPET is monotonically increasing, these three patterns are always sequential in time; Pattern 1 leads to 2, and then to 3. Exercise physiologists are particularly interested in the VO_2_ value corresponding to these transitions, i.e., the exercise thresholds. The exercise thresholds constitute important physiological markers for aerobic fitness assessment and exercise prescription. Experts visually inspect CPET results in a search for these transitions, which are announced by pronounced breaking points in specific CPET variables: The estimated lactate threshold θ_L_ (or VT1 in this manuscript, i.e., the transition from Pattern 1 to 2): identifies the highest metabolic rate not associated with acidosis or metabolic homeostasis, and it corresponds to: (a) an increase in VCO_2_ relative to VO_2_ (an increase in blood lactate concentration is associated with the increase of H^+^, which combines with HCO_3_^-^ to give an additional source of CO_2_); (b) the first disproportionate increase in VE (VE is regulated by the CO_2_ delivery to the lungs to minimize CO_2_ accumulation); (c) an increase in VEVO_2_ with no increase in VEVO_2_ (a consequence of the previous two points); (d) an increase in PetO_2_ with no consequent fall in PetCO_2_ (onset of the isocapnic period).The respiratory compensation point RCP (or VT2 in this manuscript, i.e., the transition from Pattern 2 to 3): identifies the highest metabolic rate at which homeostasis can be maintained despite a metabolic acidosis, and it corresponds to: (a) the second disproportionate increase in VE (hyperventilation relative to both VO_2_ and VCO_2_); (b) the first systematic increase in VEVCO_2_ relative to VO_2_ (a direct consequence of the previous point); (c) the first systematic decrease in PetCO_2_ (end of the isocapnic buffering period).

Experts (clinicians and exercise physiologists) detect VT1 and VT2 by the visual identification of the breaking points in the CPET variables, which are displayed on a desktop monitor. Experts typically complete the task with prior experience, personal beliefs, and the aid of computer methods [5]. Indeed, the problem of identifying the exercise thresholds by visual inspection is characterized by ambiguity, contradiction, and complexity [6]; it often requires the opinion of multiple reviewers [7], it is time- and energy-consuming and is greatly affected by intra- and inter-evaluator variability [8]. Inter-evaluator reliability has a direct effect on the reliability of the visual inspection methodology [7]. 

Most of the techniques developed for the automatic detection of VT1 and VT2 are based on regression techniques, where a function (or a set of functions) is used to fit the data and to return one of both thresholds analytically [9]. Typically, a single CPET variable is considered at the time by the algorithms currently available (e.g., VCO_2_ vs. VO_2_ for VT1 and VE vs. VCO_2_ for VT2). However, more sophisticated statistics and machine learning techniques have been applied in the context of CPET, and they might prompt a revolution in the way machines will support experts in exercise threshold detection. 

### 1.2. Regression-Generation-Explanation

In machine learning applications involving CPET analysis, it is important to make a distinction between two types of applications: First, applications that consider the statistics between tests. Hearn et al. [10] developed a feed-forward neural network (NN) for the prediction of clinical deterioration in patients with heart failure. They included the time dependence of the CPET variables by extracting features with an unsupervised classification algorithm [11]. Inbar et al. [12] adopted a support vector machine (SVM) to identify chronic heart failure and chronic obstructive pulmonary disease from CPET. Sharma et al. [13] encoded CPET and processed the output images with a convolutional neural network (CNN) for the classification of heart failure and metabolic syndrome.Second, applications that focus on the data within each test. Baralis et al. [14], for example, implemented both an SVM and a NN with a rolling window technique which considered multiple CPET variables at a time. One of their goals was the online forecast of the VO_2_ values.

The present manuscript focuses on the second category of applications and discusses the three great problems in machine learning applied to exercise threshold detection from CPET: First, the regression (or classification, or imputation) of an exercise intensity domain from the CPET variables. This challenge has already been taken by Zignoli et al., who developed a recurrent neural network (RNN) [15] and a CNN [16] to classify exercise intensity domains from a rolling window of CPET variables.Second, the generation of fake-but-realistic examples of CPET while maintaining the possibility to set the exercise thresholds a priori. To the best of the author’s knowledge, CPET data generation counts only one example in the scientific literature. Zignoli et al. [17] developed a conditional generative adversarial neural network (cGAN) to re-create a window of pre-selected CPET variables corresponding to an intensity-specific pattern.Third, the explanation of the why behind the detection of an exercise threshold. On one hand, simple regression models such as the V-slope [18] and the modified V-slope [19] can provide the expert with the physiological reason behind the disproportionate increase in VCO_2_ vs. VO_2_ and in VE vs. VCO_2_ at the exercise thresholds. However, their explanatory power comes at the expense of accuracy. On the other hand, the lack of explanatory power is a serious limitation of the use of machine learning models in the medical decision support [20,21]. Therefore, methods that could facilitate the explanation of the output of the machine learning algorithms are mostly needed [22].

At the time of writing, Oxynet (www.oxynet.net, accessed on 1 October 2022) is arguably the only framework that is considering and addressing all these challenges together, in the specific context of automatic threshold detection in CPET. The solution offered by the Oxynet project is based on a collaborative approach, where collective intelligence is used to integrate the opinions of independent and decentralized experts. Originally, the Oxynet project was conceived to identify exercise thresholds from CPET data [15,16] and to provide experts with a fast and reliable methodology to interpret new CPET. It departed from the already existing approaches in two important ways: (1) it integrated a comprehensive dataset from multiple research and evaluation centers, clinics, and universities; and (2) it provided human-level accuracy while maintaining consistent performance across populations with different aerobic fitness or across testing methodologies (long-graded vs. ramp or running vs. cycling). 

Therefore, the aim of this manuscript is to introduce and discuss the three great challenges in machine learning applied to CPET (regression, generation, and explanation) and to present the Oxynet framework, where these challenges are addressed together. 

## 2. Materials and Methods

At the core of the Oxynet algorithms, lies an extensive dataset of labelled CPET data files. These data files are being collected by independent researchers around the world. These experts are affiliated and operate in decentralized research centers, clinics, and universities. More details about the composition of the team of experts can be found at https://www.overleaf.com/read/zgsfxmvcbhkz (accessed on 1 October 2022). These files have been visually inspected, labelled (i.e., exercise thresholds are detected by the expert exercise physiologist or medical doctor), and anonymized. In line with the concept of a data crowdsourcing platform, experts can upload new labelled data files at any time to train and improve the Oxynet algorithms [16]. Since the dataset is always evolving, the composition is always updating. In this work, 575 CPET which included all the relevant CPET variables were used. This contribution only makes use of retrospective data, so consent from the participants was already collected in the original studies. Representative data from a single participant is used here to highlight the potential application of the technology, but statistics about the accuracy of the algorithms were evaluated for the entire validation dataset. 

All the models and algorithms included in the Oxynet framework have been developed in the Python (ver. 3.8) environment. All the code is public, but a large portion of the dataset is private and will be shared publicly. All the models have been embedded in a Python package called pyoxynet (https://github.com/andreazignoli/pyoxynet, accessed on 1 October 2022), which can be installed from PyPi (https://pypi.org/project/pyoxynet/, accessed on 1 October 2022). The pyoxynet package documentation is also available for consultation (https://pyoxynet.readthedocs.io/en/latest/, accessed on 1 October 2022). All the models share the same data structure (CPET variables available breath-by-breath are linearly interpolated and oversampled on a sec-by-sec time basis [15,16]) and the same basic design (they have been developed with Keras and Tensorflow (vers. 2.7.0)).

### 2.1. Regression

Oxynet implements an RNN [15] (characterized by long-short term memory elements) and a CNN [16] (characterized by convolutional layers) that takes a time-window of CPET variables and return a probability of being in an exercise domain (i.e., moderate, heavy, or severe). As previously mentioned, the ground truths (i.e., the labels), corresponding to the real exercise thresholds, were provided by experts in the field of CPET. This window is rolling from the start of the test to the end. The combination of variables considered has evolved with the versions of the model. To date, it considers 6 normalized CPET variables (between the corresponding minimum and maximum values registered during the test) such as VCO_2_/VO_2_, VE, PetO_2_, PetCO_2_, VEVO_2_, and VEVCO_2_. The rolling window is 40 s long. Therefore, the size of the input of the Oxynet neural networks adopted in the regression problem is 40 × 6, and the output is 3 × 1. Once these probabilities have been determined for the entire duration of the test, the output neuron that displays the highest values determines the exercise intensity inferred by the neural networks. Therefore, exercise thresholds are found when the probabilities of being in different exercise domains are crossing.

The current structure of the regression model consists of: (1) a convolutional layer followed by an average pooling; (2) a batch normalization, a flatten layer to adjust the dimension of the layer output, and a drop out layer; (3) a dense layer with 16 feed-forward neurons; (4) a drop out followed by another batch normalization layer; and (5) an output layer of 3 neurons. 

The CPET files in the dataset were randomly assigned to a training (80% of the files, i.e., 460 files) and a validation dataset (20% of the files, i.e., 115 files). The most straightforward method to assess the accuracy of the model estimations is to compare these estimations with the labels provided by the experts. To this, the root mean square error is computed between the values of VO_2_ corresponding to the exercise thresholds in the validation dataset. In addition, although the CPET variables are considered time series, the exercise threshold estimations are expressed in mlO_2_/min and not in time. A range of ±240 mlO_2_ is used to describe the magnitude of the errors in VT1 estimations, while a range of ±120 mlO_2_ is used to describe the magnitude of the errors in VT2 estimations. These ranges are introduced because of the intrinsic noise and variability that affects the labels, i.e., the exercise thresholds detected by the experts. Bland-Altman plots have also been used to analyze the agreement between the values estimated by the model and the values set by the experts. 

### 2.2. Generation

Oxynet implements a cGAN [17], which can generate a sample window of CPET variables starting from an input of size 53 × 1. The first 50 elements of the input tensor are random values, whilst the last three values represent the exercise intensity domain that the generative model needs to replicate. Like in the classic framework of a GAN [23], a generator was trained against a discriminator to recreate fake but realistic combinations of CPET variables with a given pattern (i.e., Pattern 1–3 mentioned in the Introduction). This last characteristic (providing a label to a generative model) is what distinguishes a classic GAN from a cGAN [24,25]. All the CPET included in the dataset (i.e., 575 files) were used in the training process.

The main building block of both the generator and the discriminator is the convolutional layer. The generator has a structure with two input heads: one for the random values of size [50 × 1] and one for the label which has size 3 × 1. After passing through multiple convolutional layers, these inputs are concatenated together and then passed to a transposed convolutional layer, which provides the last dimension of 40 × 7 (i.e., the dimension of a 40-second time window of seven CPET variables). The discriminator also has two input heads: one for the real data 40 × 7, and one for the labels 3 × 1. The inputs are concatenated together and then an output neuron provides a value of 0 (i.e., real) or 1 (i.e., fake). The output of the generator is delivered to the discriminator together with real examples, so the discriminator can learn how to separate between real and fake data. The reader interested in knowing more about the generation process is referred to the original publication [17]. 

Specifically, the cGAN implemented in Oxynet only generates a window of normalized CPET variables. To retrieve the absolute values, minimum and maximum values from the Oxynet dataset can be considered. These values are available in the pyoxynet package for a quick consultation. The seven CPET variables generated by the cGAN are as follows: VO_2_, VCO_2_, PetO_2_, PetCO_2_, VE, Rf, and HR. It is important to highlight the presence of Rf in the list of generated variables, as Rf allows the calculation of the breathing events, and it allows the transformation from the sec-by-sec time domain to the breath-by-breath domain. This feature allows for the generation of a more realistic window of CPET variables. The generator model can be called with a dedicated Python function in the pyoxynet package. There are different options that the user can use to generate a realistic CPET, e.g., the noise level, the duration of the test, the position of the exercise thresholds, and the level of aerobic fitness of the hypothetical person conducting the test.

A Flask application developed in Python can leverage the Oxynet models to engage the users in a game where they must classify real vs fake CPET examples. The app can be tested at (https://flask-service.ci6m7bo8luvmq.eu-central-1.cs.amazonlightsail.com/, accessed on 1 October 2022). The Flask app can automatically store the users’ performance and provide an instant indication of the number of tests the users are classifying correctly. Ideally, this number should be settled around ~50%, indicating an inability to correctly classify real vs. fake CPET examples.

### 2.3. Explanation

The Python library SHAP (https://github.com/slundberg/shap/, accessed on 1 October 2022) can be used to compute Shapley’s values [26] when the regressor is called. In short, in the context of complex machine learning models such as neural networks, Shapley’s values can provide indications of the specific input variable that drove the output evaluation. Three Shapley’s values (one for each regressor output) are returned for each CPET variable at each time the inference is requested. Shapley’s values can indicate the contribution of every CPET variable in pushing the model output from the base value (the average model output over the training dataset) to the model output.

More specifically, Oxynet leverages the SHAP’s DeepExplainer (which supports Keras and Tensorflow models) where a high-speed approximation algorithm for Shapley’s values is available (the implementation is close to the implementation described in [27]). Initially, pyoxynet implemented the TFLite (https://www.tensorflow.org/lite/, accessed on 1 October 2022) versions of the regression and generator models. At the time of writing, the SHAP library (https://github.com/slundberg/shap/, accessed on 1 October 2022) has not yet developed the option for TFLite to be used for the calculation of the SHAP values. Consequently, all the models included in pyoxynet are now used in their original Keras “saved model” version.

In the context of CPET exercise threshold detection, a variable with a positive Shapley’s value indicates that the variable is pushing the model output towards the corresponding exercise intensity domain. Conversely, a variable with a negative Shapley’s value indicates that the variable is pushing the model output away from the corresponding exercise intensity domain. Additionally, the magnitude of Shapley’s values for each CPET variable can give an indication of the role that every variable plays in determining the model output in every specific exercise intensity domain. Therefore, Shapley’s values can indicate why the regressor is pushing towards an exercise domain or the other, and what is the variable driving the change. 

## 3. Results

A representative example of the output of the regression models is provided in Figure 2. There are different graphical representations that can be selected to show the results to the user. The first representation (top graph, Figure 2) consists of the simple reproduction of the output neurons, which can each provide an indication of the exercise intensity domain imputed by the model. The overlap between the exercise intensity corresponding to the largest model output is determining the crossing of the relative exercise threshold. The second representation consists of a 2D map (mid graph, Figure 2), where the concept of overlapping exercise intensities is more evident. The third is the reproduction of the exercise intensity colors on the VO_2_ graph (bottom graph, Figure 2). 

A total of 115 CPET data files were used in the validation process of the regressor. Validation results are reported in Figure 3. In summary, 42% of the VT1 estimations fell within the ±120 mlO_2_ range and 73% in the ±240 mlO_2_ range. Furthermore, 73% of the VT2 estimations fell within the ±120 mlO_2_ range and 94% in the ±240 mlO_2_ range. The root mean square error between model and expert estimations was 174 and 102 mlO_2_ for VT1 and VT2, respectively. 

General statistics about the Oxynet dataset are provided in Figure 4. Typical values (average) and the dispersion of the minimum and maximum values are provided for the main CPET variables of interest. These values are used in the CPET generation problem. 

Shapley’s values are provided on a time-basis for a representative individual and CPET in Figure 5. The horizontal line with an intercept in 0 is used to better separate negative vs. positive values. This representation allows for a direct inspection of the evolution of the importance of the different variables in determining the changes in the model output. 

A comparison between the magnitude of Shapley’s values for a representative individual and CPET are provided in Figure 6. This representation allows for a direct comparison of the importance of the different variables in determining the changes in the model output. 

## 4. Discussion

The aim of this manuscript was to introduce and discuss three great challenges in the field of machine learning applied to CPET: regression, generation, and explanation. The focus of this contribution was a specific problem in CPET automatic interpretation: the automatic detection of the exercise thresholds. After a brief review of the literature on the topic, this paper listed several noticeable existing applications. 

In the Orientations towards the first Strategic Plan for Horizon Europe it is reported that: “It is a main priority for the EU to support Member States in ensuring that health care systems are effective, efficient, equitable, accessible, and resilient while remaining fiscally sustainable in the medium and long term.” The Oxynet project wants to contribute to this endeavor and promote accurate and timely decisions about exercise thresholds, ultimately reducing the costs associated with current evaluation errors and delays. 

### 4.1. Regression

Quite interestingly, human vision can be somehow approximated with computer vision [28]. Therefore, the implementation of a CNN, which is naturally conceived to find discriminatory features in patterns of variables, is most likely the closest artificial representation of what the human expert is doing when detecting the exercise thresholds in a CPET. A breaking point is just a mathematical entity for a machine learning algorithm. However, for the exercise physiologist, breaking points in CPET variables have important physiological meanings. By looking at multiple variables together, a CNN can spot differences between patterns, which are less likely to be corrupted by noise than simpler linear regression algorithms operating on single variables at a time [29,30]. Indeed, differences in exercise intensity domains have been described in the Introduction as different patterns in CPET variables instead that breaking points in time series. The author of this manuscript advocates the use of CPET variable patterns (Patterns 1–3 in the Introduction) in developing CPET machine learning applications, instead of using time series and linear regression braking point detection algorithms. The performance of the Oxynet algorithms in determining exercise thresholds from CPET is reported in published research [15,16], and summarized in Figure 3. It is found that the accuracy in the estimations made by the algorithm is compatible with that of the visual inspection, and therefore with the inter-expert variability. This is most likely due to the fact that this variability is propagated into the labels. The role of the collective intelligence on which Oxynet is based on [16] is to sift out this noise and provide an estimation, which is an average of all the contributions. 

### 4.2. Generation 

In the classic point of view, a breaking point exists when two separate lines are statistically better than a single line in fitting a set of points. The breaking point itself is found at the intersection of these two regression lines, for which the equations are available after the fitting optimization process. To generate realistic CPET variables, researchers traditionally create a set of lines with a deflection point in correspondence with the ventilatory thresholds. To these lines, noise is usually superimposed, and the original breaking point is saved for later reference. 

Experts in the field of CPET analysis are invited to try the app (https://flask-service.ci6m7bo8luvmq.eu-central-1.cs.amazonlightsail.com/, accessed on 1 October 2022) and to test their ability to separate real from fake tests. Experts are invited to provide their feedback to the author of this manuscript. The process of generating CPET data can be particularly useful in creating datasets for further supervised training of machine learning models and providing realistic alternatives to missing data points. Perhaps even more importantly, an entire dataset populated with fake tests with known pre-defined exercise thresholds could be used to create a baseline for different models to be tested. A common dataset for any model developer to test has been already suggested by Ekkekakis in 2008 [9].

### 4.3. Explanation

The process of adding an explanation to the Oxynet algorithms was inspired by the seminal work of Lundberg et al. [31], who set out to predict the risk of hypoxemia during anesthesia care and to explain the factors that led to that risk. The output of their explainable model, therefore, went beyond a single number representing the risk (e.g., the odds ratio) with a more detailed presentation of the individual factors that can sum up to provide that risk (e.g., the individual’s body mass index, tidal volume, and pulse). In many domains of health care services, general resistance to machine learning applications has been reported [20], and this has to do with the lack of explanatory power and the cognitive load associated with the interpretation of the algorithm output [21]. There is no doubt that machine learning algorithms are perceived as “black boxes” by exercise physiologists, who therefore avoid using complex algorithms. This distance between machine learning and clinical practice is known as the machine learning “chasm” [32]. However, as suggested by Lundberg et al. [33], even complex machine learning models can retain interpretability and explanatory power. Typically, understanding why a prediction was made requires limiting the complexity of the model, but the SHAP library enables explanations for models of arbitrary complexity. 

In the world of CPET, to the best of the author’s knowledge, few models with explanatory power have been presented. Jablonski et al. [34] attempted to develop an AI-assisted methodology to explain the results of a CPET machine-learning model for disease diagnosis, hence not in the context of exercise threshold detection. Jablonski et al. [34] applied a visualization technique based on gradient-weighted class activation maps [35] to their CNN, to provide interpretable results to clinicians. Their visualization techniques leveraged different transparency values for relevant portions of the signal. Portella et al. [36] set out to build an explainable model of the factors limiting the maximal effort during the CPET (pulmonary vs cardiac). In their recent work, they used several features extracted from the CPET variables, and then the SHAP library to express the contribution of each feature to the model output. 

Shapley’s values associated with the CPET variables essentially represent the change in the predicted exercise intensity domains when that variable is observed versus when the CPET variable is not considered. This change in the output prediction of the regressor when a CPET variable is considered indicates its importance for the regression (Figure 6 highlights that VE is the variable driving most of the changes on every exercise intensity domain for that specific individual). It should be noticed that variable importance does not imply a causal relationship and so does not represent an exhaustive explanation of why the exercise intensity prediction changed. However, they do enable an expert to better interpret a CPET by knowing which variable contributed to the exercise intensity predicted by the regressor. Compared to the regression or the generation, the calculation of Shapley’s values is quite expensive in terms of time and computational resources.

Explainable machine learning models for automatically estimating exercise thresholds are presented here for the first time. However, it has yet to be established whether Shapley’s values and their representations can help final-product end-users such as clinicians, exercise physiologists, and sports scientists in their practice. 

### 4.4. Practical Applications

The algorithms presented in the Oxynet framework can facilitate CPET data interpretation and analysis. For example, it is believed that the automatic identification of the exercise threshold could save time for the exercise physiologist. This time could be devoted to building a better and more insightful human relationship with the individual/patient/athlete. This is arguably a hope for every machine learning application in health care [37]. To date, the Oxynet algorithms are used by a small number of users, progressively increasing over time. Among these users are exercise physiologists and medical doctors, who make use of Oxynet to save time and energy, or because they need an additional objective and independent opinion about a CPET test. The users report that the algorithm is fast and can clearly provide the estimations (such as those reported in Figure 2) within seconds. 

In some settings, requiring an exhaustion test is highly demanding. This is especially true for individuals who are not used to high-intensity efforts. Therefore, having the possibility to retrieve VT2 without going to exhaustion is highly useful. Indeed, the regression algorithm presented here could be adapted for real time applications, where the crossing of the heavy-to-severe exercise intensity domain is notified by a warning. These applications could run on portable or desktop metabolimeters. The generation algorithm could be implemented to train the next generation of exercise physiologists or medical doctors with gold-standard custom-created data files, or to generate a dataset for developing and training new algorithms for data interpretation. CPET software could benefit from the Oxynet models, and the explanations provided with the Shapley’s values to improve graphical interfaces and provide additional insights about the inference results. 

### 4.5. Final Considerations

This manuscript discussed a particular application of machine learning algorithms applied to CPET data analysis and interpretation: the automatic detection of the exercise thresholds. In this specific context, the innovative aspects of this contribution are at least two: (1) regression, generation, and explanation problems have been discussed under a single framework; and (2) the explainable machine learning models are discussed here for the first time in this context. It is important to highlight that the Oxynet project [16] is not an isolated effort. The number of AI applications related to CPET has been rapidly increasing in the last decade. Most applications such as those considering the global information that a single test can provide [10], or those using encoded CPET as images [13] as well as time series [14,38], are perfectly complementary to this research. The energy and time savings, and the improved level of objectivity that these applications can provide, should not be underestimated by those operating in the CPET related industries, i.e., companies producing and commercializing CPET hardware (both desktop and portable devices) and software, or final-product end-users such as clinicians, exercise physiologists, and sports scientists.

## 5. Conclusions

The Oxynet framework can deal with the regression, generation, and explanation of exercise thresholds in CPET data. The Oxynet algorithms’ high readiness level is ensured by the presence of an open-source Python package (pyoxynet). The introduction of explanatory values should allow accurate, but traditionally hard-to-interpret, models to be used while still providing intuitive explanations of what led to an estimated exercise intensity probability. Ultimately, Oxynet wants to play its small role in improving universal access to good quality health care. The models provided with Oxynet could be embedded in portable and desktop CPET devices and extend the outreach of exercise physiologists beyond their local clinics and laboratories. 

## Figures and Tables

**Figure 1 sensors-23-00826-f001:**
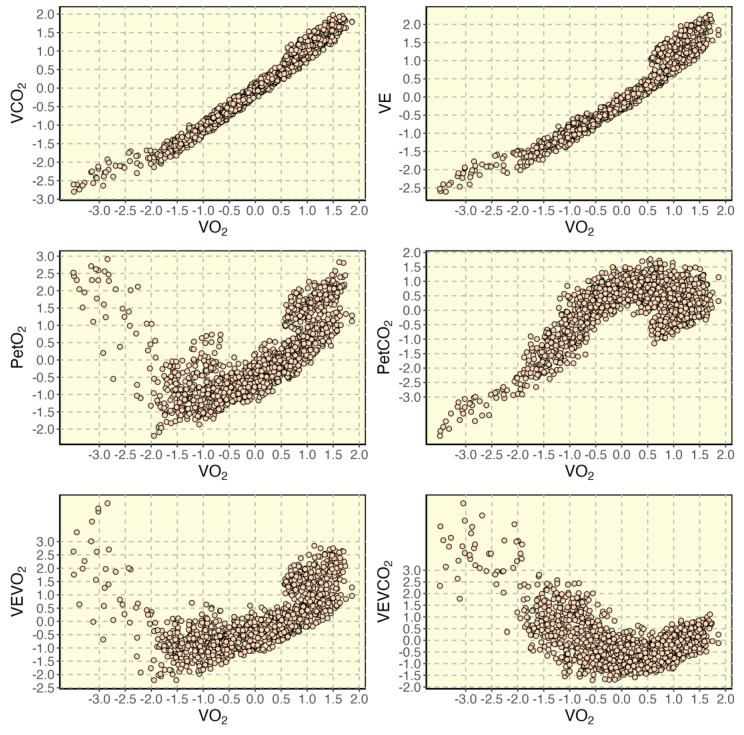
Main ventilatory variables collected during a CPET (standardized units) for a representative individual: exhaled CO_2_ (VCO_2_), ventilation (VE), end-tidal O_2_ (PetO_2_) and CO_2_ (PetCO_2_), ventilatory equivalents (i.e., VEVO_2_ and VEVCO_2_) are plotted relative to O_2_ uptake (VO_2_).

**Figure 2 sensors-23-00826-f002:**
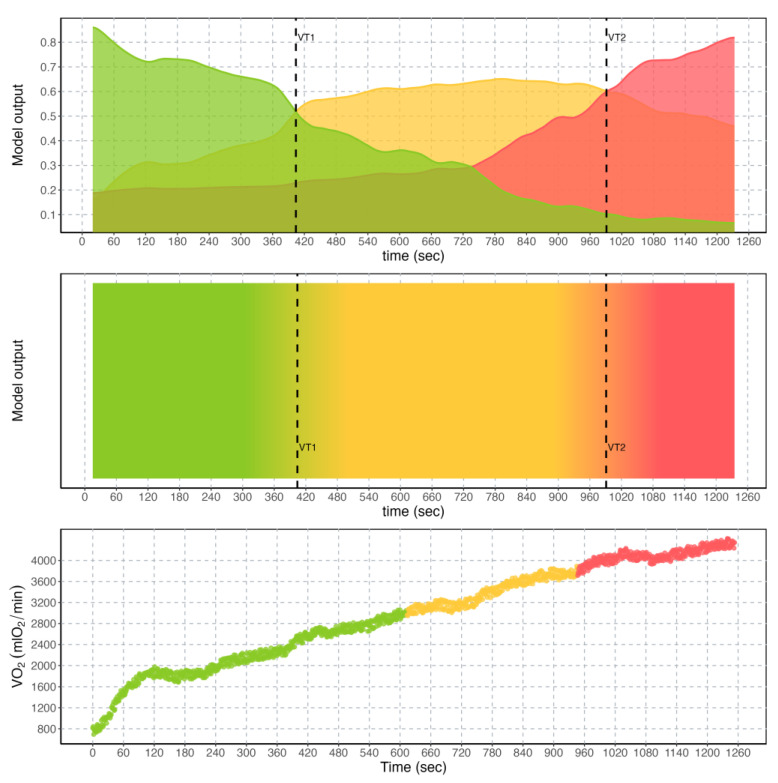
Output of the regressor model currently implemented in Oxynet for a representative individual. (**Top**): output neurons of the convolutional neural network, corresponding to the three exercise intensity domains (i.e., moderate (green), heavy (yellow), and severe (red)) relative to time in seconds. Exercise thresholds VT1 and VT2 are found where the output of the neural network overlap. (**Middle**): 2D colored map of the output neurons relative to time in seconds. (**Bottom**): typical behavior of the O_2_ uptake (VO_2_) in mlO_2_/min relative to time in seconds for a representative individual, where colors are used to highlight different exercise intensity domains.

**Figure 3 sensors-23-00826-f003:**
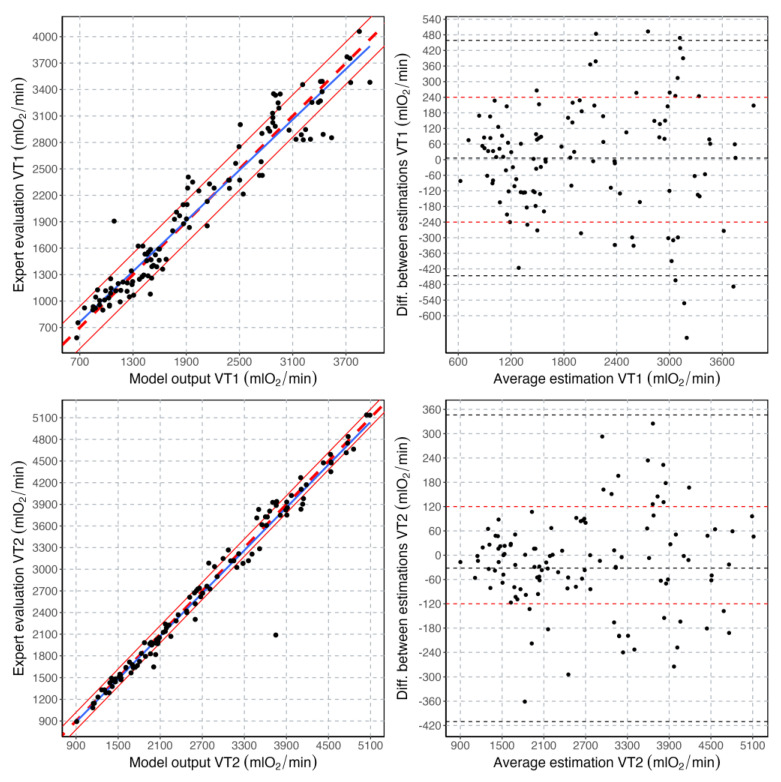
Correlation plots and Bland-Altman plots for the exercise threshold values estimated with the machine learning model and those estimated by the experts. In the correlation plots, the blue line is the regression line, the thick dashed red line is the line with a slope of 1, and the thin continuous lines indicate the ±240 and ±120 mlO_2_ ranges for VT1 and VT2, respectively. In the Bland-Altman plots, the average value and the limits of agreement are indicated with a dashed black line. The ±240 and ±120 mlO_2_ ranges are also highlighted by horizontal dashed red lines for VT1 and VT2, respectively.

**Figure 4 sensors-23-00826-f004:**
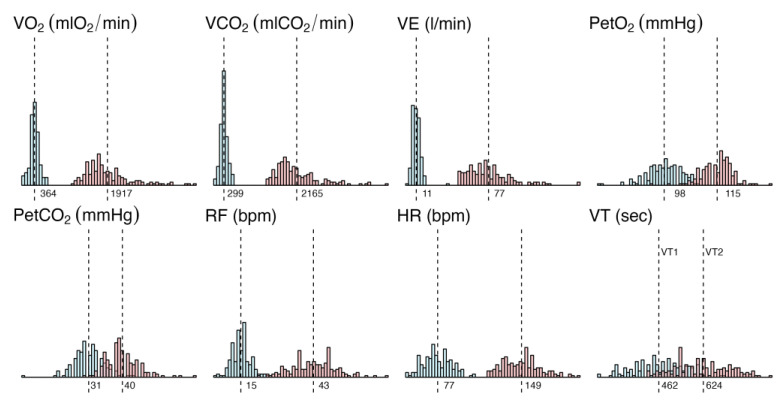
Distribution of the minimum and maximum values (and their corresponding average) of the main variables across the entire Oxynet dataset: O_2_ uptake (VO_2_) in mlO_2_/min, exhaled CO_2_ (VCO_2_) in mlCO_2_/min, ventilation (VE) in l/min, end-tidal O_2_ (PetO_2_) and CO_2_ (PetCO_2_) in mmHg, respiratory frequency (RF) in bpm, heart rate (HR) in bpm and exercise thresholds (VT1 and VT2) in seconds from the beginning of the exercise.

**Figure 5 sensors-23-00826-f005:**
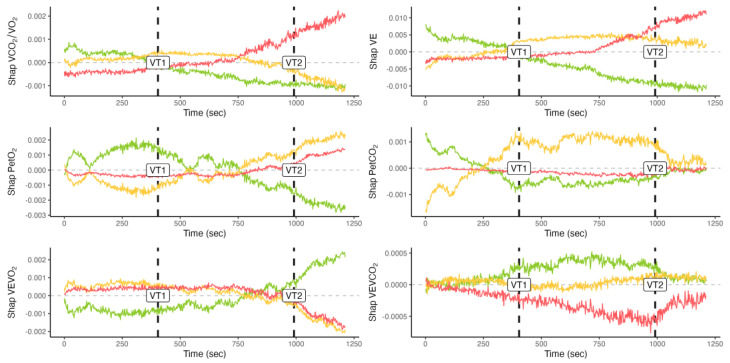
Shapley’s values for a representative individual and test. Colors are used to separate exercise intensity domains, i.e., moderate (green), heavy (yellow), and severe (red). Time is provided in seconds from the beginning of the test. Exercise thresholds detected by the neural network are also highlighted (VT1 and VT2).

**Figure 6 sensors-23-00826-f006:**
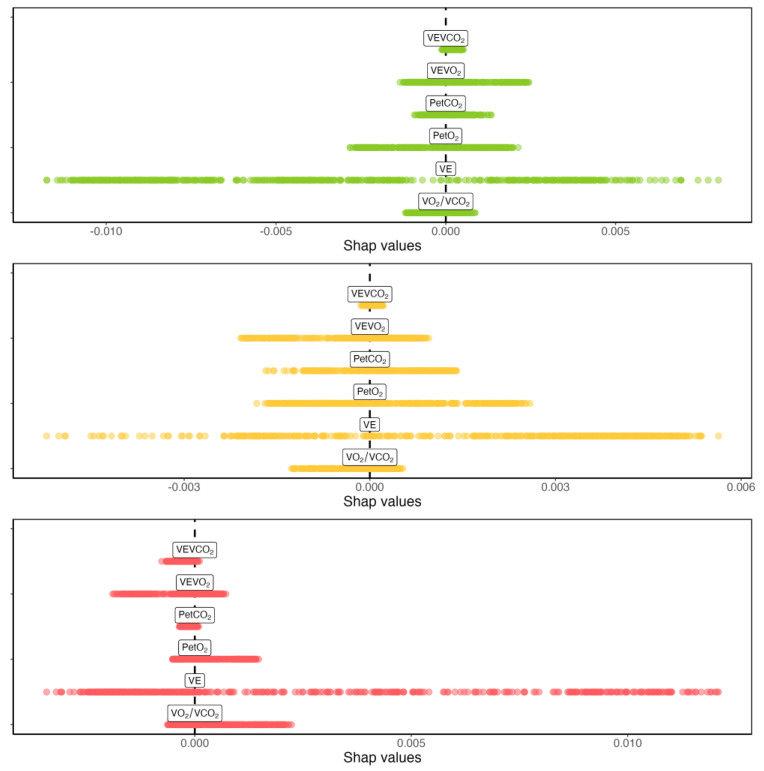
Comparison highlighting the magnitude of Shapley’s values for a representative individual and test. Colors are used to separate exercise intensity domains, i.e., moderate (green), heavy (yellow), and severe (red).

## Data Availability

All the relevant information and the code used to generate the results presented in this manuscript are available in the project repository: https://github.com/andreazignoli/pyoxynet (accessed on 1 October 2022).

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
