# Peer review of "Machine Learning Models for the Automatic Detection of Exercise Thresholds in Cardiopulmonary Exercising Tests: From Regression to Generation to Explanation"

_sensors, 2023, doi:10.3390/s23020826_

Round 1
Reviewer 1 Report
Dear Author,
Thank you for the insight into this area that really helpful for me as an eye-opener.
I would like to suggest from my own thoughts to be considered for the improvement of this manuscript:
Line 142… The latest details about the composition and the quality of the Oxynet dataset are provided in [24]…. to mention where the actual “location” of the details
To highlight this study is a case report because involves one representative individual and for that, consent from that particular individual may be needed.
To include validation testing to verify the results.
Author Response
R1.1 Line 142… The latest details about the composition and the quality of the Oxynet dataset are provided in [24]…. to mention where the actual “location” of the details
Thank you. Details about the exact location where to find details about the dataset have been included with an external link.
R1.2 To highlight this study is a case report because involves one representative individual and for that, consent from that individual may be needed.
Thank you. This study makes use of data retrospectively, so consent for this specific participant was already collected for the original study. Representative data from a single participant was used here to highlight the potential application of the technology, but statistics about the accuracy of the algorithms were evaluated for the entire validation dataset, which included a high number of individuals. These details were included in the manuscript.
R1.3 To include validation testing to verify the results.
Thank you. Validation testing details have been included.
Reviewer 2 Report
This paper introduces machine learning models for the automatic detection of exercise thresholds in cardiopulmonary exercising tests. I have some suggestions on this paper.
1-The abstract can be revised to include more details of the proposed methodology and the results.
2-Please check the full manuscript and pay attention to the subscripts, such as O2 and CO2.
3-2. Materials and Methods: This section is too brief. It should be improved and added the theory and the steps.
4-It is mentioned in the manuscript that the given method can be used for hardware (both desktop and portable devices) and software, or final-product end-users such as clinicians, exercise physiologists, and sports scientists. It is better to give an example of practical application.
Author Response
R2.1 The abstract can be revised to include more details of the proposed methodology and the results.
Thank you. More details about the methodology and the results have been included in the abstract.
R2.2 Please check the full manuscript and pay attention to the subscripts, such as O2 and CO2.
Thanks for your comment. Subscripts have been checked and amended.
R2.3 Materials and Methods: This section is too brief. It should be improved and added the theory and the steps.
Thank you. More details have been included in the Materials and Methods sections, especially in the validation aspects of the regression algorithm.
R2.4 It is mentioned in the manuscript that the given method can be used for hardware (both desktop and portable devices) and software, or final-product end-users such as clinicians, exercise physiologists, and sports scientists. It is better to give an example of a practical application.
Thank you. This is a pertinent comment. Additional examples of possible practical applications have been included in a new section.
Reviewer 3 Report
The author has explored machine learning models for automatically detecting exercise thresholds in CPET. Please see below the comments on the same:
1. I am curious if the model provides specific limitations to guided exercise.
2. How easy is it to interpret the results from this model for clinicians? The author claims that the model could be easily embedded. It would be helpful if the results section profoundly highlights the advantages and straightforward interpretation.
3. Please include the units during the first mention of all the terminology and variables.
4. Abstract, and materials can be improved. Please add more details for easy flow.
Best!
Author Response
R3.1 I am curious if the model provides specific limitations to guided exercise.
Thank you for your comment. We assume that with "guided exercise" we mean a prescribed exercise with a priori established exercise intensities. If the exercise thresholds can be accurately estimated, then they can be used in exercise prescription and performance assessment. This is one of the main advantages of conducting CPET gold-standard assessment for the individual's aerobic fitness. A brief sentence was added in the introduction to highlight the usefulness of exercise thresholds.
R3.2 How easy is it to interpret the results from this model for clinicians? The author claims that the model could be easily embedded. It would be helpful if the results section profoundly highlights the advantages and straightforward interpretation.
Thank you for your comment. We are not entirely sure if Shapley's values will be easily digested by clinicians and experts in the field of CPET. However, experts have been asking for explainable models for exercise thresholds, so this is the first attempt in that direction. Conversely, for the regression algorithm, estimated exercise thresholds have a straightforward interpretation, since they are provided in terms of VO2, and they can be received by experts within seconds. Additional comments have been included in the manuscript.
R3.3 Please include the units during the first mention of all the terminology and variables.
Thank you. Measurement units have been included.
R3.4 Abstract, and materials can be improved. Please add more details for easy flow.
Thank you. According to this comment and to the first and third comments of the second reviewer, more details about the methodology and the results have been included in the manuscript.
Round 2
Reviewer 2 Report
The paper has been carefully revised. But There are several other problems.
1- The affiliation: Department of Industrial Engineering. Which university?
2- Fig. 1, Fig. 3 & Fig. 6: Please check the subscripts.
3- Fig. 2 & Fig. 5: Please mark the unit of time.
Author Response
Thank you again for taking the time to revise this manuscript.
R2.1. 1- The affiliation: Department of Industrial Engineering. Which university?
Checked and fixed. Thanks.
R2.2. 2- Fig. 1, Fig. 3 & Fig. 6: Please check the subscripts.
Checked and fixed. Thanks.
R2.3. 3- Fig. 2 & Fig. 5: Please mark the unit of time.
Checked and fixed. Thanks.